# Human Bone Marrow-Derived Mesenchymal Stem Cell Applications in Neurodegenerative Disease Treatment and Integrated Omics Analysis for Successful Stem Cell Therapy

**DOI:** 10.3390/bioengineering10050621

**Published:** 2023-05-22

**Authors:** Seok Gi Kim, Nimisha Pradeep George, Ji Su Hwang, Seokho Park, Myeong Ok Kim, Soo Hwan Lee, Gwang Lee

**Affiliations:** 1Department of Molecular Science and Technology, Ajou University, 206 World Cup-ro, Suwon 16499, Republic of Korea; rlatjrrl9977@ajou.ac.kr (S.G.K.); nimishapgeorge@ajou.ac.kr (N.P.G.); js3004@ajou.ac.kr (J.S.H.); 2Department of Physiology, Ajou University School of Medicine, 206 World Cup-ro, Suwon 16499, Republic of Korea; gdj3315@ajou.ac.kr (S.P.); shwanlee@ajou.ac.kr (S.H.L.); 3Department of Biomedical Science, Graduate School of Ajou University, 206 World Cup-ro, Suwon 16499, Republic of Korea; 4Division of Life Science and Applied Life Science (BK21 FOUR), College of Natural Sciences, Gyeongsang National University, Jinju 52828, Republic of Korea; mokim@gnu.ac.kr

**Keywords:** neurodegenerative diseases, stem cell therapy, human bone-marrow-derived mesenchymal stem cells, integrated omics, stemness

## Abstract

Neurodegenerative diseases (NDDs), which are chronic and progressive diseases, are a growing health concern. Among the therapeutic methods, stem-cell-based therapy is an attractive approach to NDD treatment owing to stem cells’ characteristics such as their angiogenic ability, anti-inflammatory, paracrine, and anti-apoptotic effects, and homing ability to the damaged brain region. Human bone-marrow-derived mesenchymal stem cells (hBM-MSCs) are attractive NDD therapeutic agents owing to their widespread availability, easy attainability and in vitro manipulation and the lack of ethical issues. Ex vivo hBM-MSC expansion before transplantation is essential because of the low cell numbers in bone marrow aspirates. However, hBM-MSC quality decreases over time after detachment from culture dishes, and the ability of hBM-MSCs to differentiate after detachment from culture dishes remains poorly understood. Conventional analysis of hBM-MSCs characteristics before transplantation into the brain has several limitations. However, omics analyses provide more comprehensive molecular profiling of multifactorial biological systems. Omics and machine learning approaches can handle big data and provide more detailed characterization of hBM-MSCs. Here, we provide a brief review on the application of hBM-MSCs in the treatment of NDDs and an overview of integrated omics analysis of the quality and differentiation ability of hBM-MSCs detached from culture dishes for successful stem cell therapy.

## 1. Introduction

Neurodegenerative diseases (NDDs) are caused by the progressive degeneration of the structure and function of neurons and glial cells in the central and peripheral nervous systems [1,2]. NDDs can be classified according to their causes and symptoms [3]. Alzheimer’s disease (AD) and Parkinson’s disease (PD) are among the most common neurodegenerative disorders. AD shows widespread degeneration of several types of neurons, whereas PD shows selective loss of a specific cell population, such as dopaminergic neurons. Amyotrophic lateral sclerosis (ALS), commonly called Lou Gehrig’s disease, shows selective degeneration of the upper and lower motor neurons [4] and has been associated with genetic mutations in the enzyme Cu/Zn superoxide dismutase 1 (SOD1) [5,6]. Huntington’s disease (HD) is a rare genetic disorder caused by a mutation in the huntingtin gene that causes movement abnormalities and cognitive impairment [7]. As the human lifespan increases, the social burden of NDDs also increases [8]. Due to the incurable and debilitating nature of these conditions, there are several limitations in treating NDDs with conventional methods. Therefore, there is a need for new treatment approaches.

Transplanted stem cells exert paracrine effects on damaged neuronal cells and replace lost neurons or glial cells [9,10]. Thus, stem-cell-based approaches are potential therapies in myocardial infarction [11], ischemic diseases [12], spinal cord injury [13], and NDDs, such as AD (NCT01547689 and NCT02833792) [4], PD [14], and multiple system atrophy [15]. Furthermore, HD could be treated with genetically engineered mesenchymal stem cells (MSCs) overexpressing brain-derived neurotrophic factor (BDNF) (NCT01937923) [16]. ALS is another NDD that could be treated using stem cells [4]. There are several MSC types, such as adipose-derived MSCs, umbilical-cord-derived MSCs, tonsil-derived MSCs [17], dental-pulp-derived MSCs (DPSCs) [18], and bone-marrow-derived MSCs (BM-MSCs). Among them, BM-MSCs, first identified in guinea pig bone marrow in 1970 [19], are attractive therapeutic agents because of their widespread availability, easy attainability, and in vitro manipulation and the absence of ethical concerns [20,21,22]. For these reasons, BM-MSCs are extensively used in clinical applications and therefore the biological characteristics and clinical effects of BM-MSCs are far more well established than MSCs obtained from other sources [23]. In addition, the expression level of HLA class I or II is low in BM-MSCs, avoiding activation of allogenic lymphocytes [24], and BM-MSCs are safe from tumorigenicity after transplantation than other stem cells, including induced pluripotent stem cells (iPSCs) and neural stem cells [25,26]. BM-MSCs exhibit homing to injured sites [27,28,29,30], angiogenic ability [31,32], anti-inflammatory effects [33,34], differentiation capability [35,36], anti-apoptotic properties [37,38], and trophic factor secretion [39,40,41,42]. Furthermore, BM-MSCs have been engineered to express tropic factors, such as BDNF [33,41,43], glial-cell-line-derived neurotrophic factor (GDNF) [44,45,46], and nerve growth factor (NGF) [47,48] to improve therapeutic paracrine effects for NDD treatment. Thus, BM-MSC-based cell therapy is a promising approach for treating NDDs [49,50].

Because MSCs can accelerate tissue regeneration, they are excellent candidates for tissue engineering. Vacanti and Langer define tissue engineering as an interdisciplinary field in which engineering and biological sciences are applied together to develop biological substitutes that can restore, maintain, or enhance tissue function [51]. Tissue engineering facilitates autologous MSCs’ transplantation by seeding the patient’s cells onto a biodegradable scaffold that forms a specific tissue that can then be used to repair injuries [52]. Despite all this, MSC-based tissue engineering faces some challenges, such as the low survival rates of MSCs and the uncertainty of MSC differentiation after infusion. To address these shortcomings, biomaterials are used together with MSCs to maintain their viability, serve as a substrate for cell adhesion, induce differentiation into specific target cells, and as a mechanical tool for tissue regeneration. Biomaterials have demonstrated excellent biocompatibility, provide a suitable cellular microenvironment for MSCs, and are effective and easy to administer. As biomaterials are diverse in their physical, chemical, mechanical, and biological properties, they can contribute to tissue regeneration in different types of injuries [53]. Although several biomaterials have been developed, hydrogels, nanofibers, carbon-based nanomaterials, and cell-free scaffolds have emerged as the frontrunners. For example, the 3D structure of hydrogels supports the proliferation of cells while acting as a barrier to harmful factors [54]. Yan et al. investigated collagen–chitosan scaffolds with BM-MSCs as a therapeutic strategy for traumatic brain injury. Collagen scaffolds with human MSCs have been shown to improve spatial learning and sensorimotor functions, while chitosan serves as a neuroprotector. They reported that these scaffolds exhibited low immunogenicity, good biocompatibility, and therapeutic effects such as neurological, behavioural, and cognitive recovery [55].

In clinical applications, the quality of hBM-MSCs should be analyzed after cell detachment from the culture dish [12,15,56], because cell quality is an essential factor for therapeutic effects. However, there are limitations to the traditional methods used to evaluate the quality and freshness of cells prior to transplantation of hBM-MSCs into humans, such as trypan blue staining or fluorescence-activated cell sorting (FACS) [56,57]. Regardless, it is possible to increase the treatment effect by determining the exact cell condition using various big data analyses, including omics and bioinformatic analyses of the quality and differentiation ability of hBM-MSCs over time after cell detachment from the culture dish. In this review, the transcriptome and metabolome, analyzed using microarray and gas chromatography-mass spectrometry (GC-MS), respectively, were integrated to evaluate the condition of hBM-MSCs. Integrated omics analysis of hBM-MSCs showed increases in reactive oxygen species (ROS) production, lipid peroxidation, and cell damage, leading to the loss of cell quality and differentiation ability as the phosphate-buffered saline (PBS) storage time increased (Figure 1).

Omics-based approaches are actively applied and developed to reflect diverse and complementary biological phenotypes and elucidate precise molecular mechanisms as the availability of high-throughput data technologies increases in the biological and medical fields [59]. These have been beneficial in identifying biomarkers for the diagnosis of various diseases. Nevertheless, most omics analyses have been limited to a single dataset, which is accompanied by difficulty in reflecting the actual phenotype [60]. In addition, the datasets used for computational analysis have advanced from structured one to big data with various unstructured and semi-structured characteristics, and the relationship between omics data is expected to become more complex [61]. Artificial intelligence (AI) is increasingly essential in big data mining, including the biological and medical fields [62,63]. Among AI, machine learning and deep learning approaches exert tremendous power in processing and modelling vast and diverse omics data [64]. Integrated omics analysis aims to utilize big data, machine learning, and systematic algorithms to obtain patterns between data and make more accurate predictions [65,66]. Convergence with computational science is necessary for this type of analysis [67].

The current review has two sections. In the first section, we provide an overview of studies on the application of hBM-MSCs to the treatment of various NDDs, and in the second section, we provide an integrated omics analysis of the quality and differentiation ability of hBM-MSCs according to PBS storage time for successful stem cell therapy.

## 2. Application of hBM-MSCs for the Treatment of Various Neurodegenerative Diseases

Neurodegenerative diseases are characterized by the selective dysfunction and progressive loss of neurons, glial cells, and neural networks in the brain and spinal cord. Synaptic dysfunction, neuronal loss, proteasome dysfunction, and the aggregation of misfolded proteins are common NDD features. NDDs affect multiple facets of function in humans, thereby limiting their ability to perform even the most basic tasks [4,68,69].

Stem cells are crucial for the development, growth, and repair of various tissues and organs. In addition, stem cells have provided breakthroughs across all fields of research and medicine owing to their multipotency and self-renewal properties. MSCs have demonstrated numerous neuroprotective effects such as decreased apoptosis, reduced ROS generation, and the promotion of neuronal growth. In particular, hBM-MSC transplantation has been shown to improve clinical outcomes, decrease cerebral atrophy, and enhance patient performance [4,70,71]. In this section, we briefly review the neuroprotective effects of clinically applied hBM-MSCs on some common NDDs.

### 2.1. AD

AD is an NDD associated with a progressive decline in cognitive and memory functions. The pathological features of AD include the aggregation of amyloid beta peptides (Aβ), forming amyloid plaques, intracellular neurofibrillary tangles, and hyperphosphorylated tau and leading to neuronal death. Continual build-up of Aβ activates microglia, thus accelerating neuronal loss, cognitive decline, tau pathology, and the secretion of proinflammatory cytokines. These factors induce synaptic deficits in the hippocampus, leading to cognitive impairment and memory decline [50,68,71,72]. Conventional AD treatments consist of two types of pharmacological therapies. The first includes the use of cholinesterase inhibitors to relieve physical symptoms by increasing the levels of the neurotransmitter acetylcholine. The second type of therapy uses memantine, a drug that improves symptoms by inhibiting N-methyl-D-aspartate (NMDA) receptors. Although several drugs and natural compounds are available for the treatment of AD, drugs that can prevent or delay the progression of AD are yet to be discovered [72].

Transplanted hBM-MSCs can differentiate into neurons, produce neurotrophic factors such as BDNF and NGF, and inhibit Aβ- and tau-related cell death [71]. Numerous studies have explored the neuroprotective effects of BM-MSCs in AD mouse models. Lee et al. achieved a significant reduction in oxidative stress, improvement in cognitive function, and mitigation of Aβ-induced neuronal injury both in vitro and in vivo after co-culturing BM-MSCs with hippocampal neurons stimulated by Aβ [73]. In addition, hMB-MSC-derived vesicles alleviated cognitive decline, reduced the number of intracellular plaques, decreased chronic inflammation, and delayed AD pathogenesis in a preclinical mouse model [74]. BM-MSC-derived exosomes have also been shown to ameliorate cognitive damage by secreting miRNAs capable of enhancing neuronal plasticity, promoting cell survival and synaptogenesis, and suppressing inflammation [75]. Collectively, these studies highlight the multiple advantages of BM-MSCs as a therapeutic agent for AD.

### 2.2. PD

PD is the most common synucleinopathy characterized by the loss of dopaminergic neurons in the substantia nigra pars compacta (SNpc) and the accumulation of α-synuclein in Lewy bodies, causing tremors, bradykinesia, and cognitive dysfunction [76]. Dopamine (DA) is a neurotransmitter that transmits information between the SNpc and other parts of the brain, thereby controlling the body’s voluntary movements. Mitochondrial dysfunction, excessive ROS generation, and impairment of the ubiquitin-proteasome system are involved in DA neuronal degeneration [77]. A typical therapy for PD involves treatment with the DA precursor L-3,4-dihydroxyphenylalanine (L-Dopa), which can produce adverse effects, such as non-responsiveness and abnormal uncontrollable movements or dyskinesia, upon long-term use. Furthermore, this form of therapy focuses only on alleviating symptoms instead of resolving the primary cause of the disease, thereby permitting disease progression [78,79].

MSC-based cell therapies offer diverse options for the treatment of PD. One case in point involves the implication of a defective autophagy system as a plausible cause of PD. MSCs have been reported to display α-synuclein clearance, the regulation of autophagy lysosomal activity, the activation of autophagy signalling and immunomodulatory effects, such as penetrating injured sites, releasing numerous growth factors, and attenuating inflammation [79]. Another study investigating neural-induced hBM-MSCs (NI-hBMSCs) demonstrated increased cell survival, stabilization of α-synuclein monomers, and promotion of neurogenesis after treatment with NI-hBMSCs [80]. Clinical trials in which hBM-MSCs from healthy donors were intravenously administered to patients with PD have shown promising results. The participants in the study exhibited post-infusion changes in motor and non-motor symptoms that lasted until the end of the study period [81]. Based on the results from these and several other studies on patients with PD and animal models, the therapeutic benefits of hBM-MSCs over conventional treatments are evident.

### 2.3. ALS

ALS is a gradual, fatal, paralytic NDD characterized by the degeneration of the upper and lower motor neurons. ALS causes weakness and atrophy of the muscles of the limbs, chest, neck, and oropharyngeal area and eventually death due to respiratory failure. As there are only two drugs currently approved for ALS treatment, there is an urgent need for different treatment options [82,83]. MSCs are being explored as a treatment option for ALS because they produce and release neurotrophins, which are proteins that induce the survival, development, and function of neurons. Transplantation of hMSCs has also been reported to mitigate neuroinflammation, improve motor execution, and enhance the bioenergetics of recipient cells [84,85,86].

Clinical trials using hBM-MSCs as therapeutic agents for ALS have shown favourable outcomes after hBM-MSC administration. In one open-label phase I trial, TGF-β and IL-10 levels were elevated following hBM-MSCs administration. TGF-β, a growth factor involved in various aspects of neuron development and function, has been found to be reduced in ALS patients and inversely correlated to disease progression [87]. Another open-label study conducted to evaluate the safety and efficacy of autologous hBM-MSCs via intrathecal and intravenous routes in ALS patients demonstrated a temporary decline in ALS progression after a single dose of hBM-MSCs [88]. To maximize the capability of hBM-MSCs to treat ALS, additional studies are needed on the effective delivery of MSCs to patients, the effectiveness of MSCs expressing diverse growth factors, and their clinical significance [89].

### 2.4. HD

HD is an autosomal dominant neurodegenerative disease caused by the loss of gamma-aminobutyric acidergic (GABAergic) medium spiny neurons in the striatum. This neuronal loss stems from the expansion of the cytosine–adenine–guanine (CAG) repeat within exon 1 of the huntingtin (*htt*) gene, which leads to the formation of a malfunctioning mutant HTT protein [90]. The clinical manifestations of HD include chorea, psychiatric symptoms, and cognitive impairment. The reduced availability of neurotrophic factors, such as NGF, BDNF, and neurotrophin-3 (NT-3), contributes to neurodegeneration and therefore, these are considered potential therapeutic agents for HD [91].

In a mouse model of HD using transplanted hBM-MSCs, intrastriatally transplanted hBM-MSCs not only successfully survived and differentiated but also reduced motor function impairment, increased neurogenesis, and boosted animal survival and cell differentiation [92]. Given that levels of neurotrophic factors (NTFs) are reduced in patients with HD, NTF-based therapies are potential strategies for the discovery of new treatment options for HD. Since BDNF has a short half-life, which limits effective delivery strategies for NDDs, genetically engineered hBM-MSCs that deliver BDNF (MSC/BDNF) have advantages of delivering BDNF to the striate and MSC-secreted factor supplementation. Pollock et al. reported a significant surge in neurogenesis, an increased lifespan, and decreased spinal atrophy in mice transplanted with MSCs/BDNF [93]. Moreover, hBM-MSCs induced to differentiate into NTF-secreting cells (NTF^+^) exhibit therapeutic properties and attenuate neurotoxicity [94]. Although the therapeutic ability of hBM-MSCs in the treatment of HD has been established, additional studies to accurately determine the administration time, dose, and frequency of cells, as well as long-term toxicology studies, should be conducted.

## 3. Analysis of the Quality and Differentiation Ability of hBM-MSCs

To date, many studies on NDDs have shown different features of neurodegeneration, such as cell viability reduction, genetic mutations, gene expression alterations, and cellular function impairment [95,96,97]. To understand the different cellular processes in NDDs, cell health has been evaluated using several methods, including morphological analysis, viability assays, metabolic assays, and gene expression analysis [98,99,100], because cell quality directly affects cell functionality [101,102].

Similarly, because the quality and differentiation potential of stem cells are crucial for their use in NDD therapy, the condition and properties of stem cells should be assessed prior to use [103]. Moreover, BM-MSCs show a loss of stemness when maintained under certain conditions [104,105]. Stolzing et al. reported that the overall hBM-MSC fitness decreased with age, showing an increase in ROS, p21, and p53 levels [104]. These deteriorating features were also observed during in vitro ageing. Geissler et al. showed that progenitor characteristics were lost, and genes related to cell differentiation, focal adhesion organisation, cytoskeleton turnover, and mitochondrial function were downregulated during long-term in vitro expansion of MSCs [105].

Meanwhile, there are some studies suggesting that gender may affect the efficiency of BM-MSC therapy [106,107]. Sammour et al. reported that female BM-MSCs have more therapeutic effects than male BM-MSCs by showing greater pro-angiogenic and anti-inflammatory effects in mice models [106]. In addition, Crisostomo et al. demonstrated that female BM-MSCs showed lower apoptosis, TNF and IL-6 production, and higher VEGF expression upon stress activation than males, due to their inherent resistance to TNFR1 activation [107]. However, one study reported that in vitro mesodermal differential capacity of hBM-MSCs is not highly related to the donor gender [108]. Although donor gender seems to play a role in the therapeutic effects of BM-MSCs, this is still not clearly elucidated and further studies are required to clarify the effect of gender on BM-MSCs.

Even though MSCs have several advantages, including self-regeneration ability, anti-inflammatory and immunomodulatory effects, and multi-lineage differentiation ability [109,110], keeping them fresh and healthy to maintain their beneficial properties should also be considered.

### 3.1. Optimizing hBM-MSCs While Preserving Cell Quality

As mentioned above, it is imperative to preserve the quality and properties of stem cells, which significantly affect the achievement of many therapeutic cells during ex vivo expansion [56]. Furthermore, before clinical use, MSCs should be isolated and expanded in vitro until they reach the appropriate cell number, owing to the low frequency of 0.001–0.01% of the total mononucleated cells [111]. hBM-MSCs are detached from the culture dish and kept in a largely different environment from the original one, which can diminish their valuable properties [112,113,114]. Hence, maintaining the quality of hBM-MSCs is crucial when using in NDDs. In this section, we summarize several methods of improving the efficiency of BM-MSC therapy while preserving cell stemness.

Donor age is a well-known factor that should be considered during the transplantation process. Many studies have reported that donor age is closely related to negative effects on MSC proliferation and multipotency [115,116,117]. Zaim et al. investigated the effects of donor age and long-term culture on the morphology, characteristics, and capacity of hBM-MSCs to proliferate and differentiate into adipogenic, chondrogenic, osteogenic, and neurogenic lineages [118]. They found that hBM-MSCs gradually reduced their proliferation rate and lost their typical morphology in an age- and passage-dependent manner. In another study, the older donor group showed lower concentrations of colony forming unit fibroblasts (CFU-Fs) and shorter telomere lengths during in vitro expansion than the young donor group, indicating that the ageing of MSCs can reduce their therapeutic characteristics [119].

Moreover, avoiding oxidative stress is effective in maintaining cell freshness. For example, Shin et al. reported that hBM-MSCs trypsinized and maintained in PBS showed a significant loss of freshness and viability over time [57]. This reduction in freshness is accompanied by several phenotypes, such as increased peroxidation of membranes and intracellular vacuoles, in a PBS storage time-dependent manner [58]. In addition, transcriptomic analysis of hBM-MSCs stored in PBS for 12 h predicted increased ROS generation and lipid peroxidation and decreased cell viability [56]. Regarding ROS production, Lee et al. suggested that treatment with N-acetyl-L-cysteine (NAC) and glutathione (GSH) maintained the quality of PBS-stored hBM-MSCs by reducing ROS and lipid peroxidation [56]. Indeed, one study showed that the pre-conditioning of BM-MSCs with NAC led to lower apoptosis and higher survival against oxidative stress by increasing GSH levels and helped bone regeneration after transplantation in rats [120].

Lastly, treatment with growth factors, such as fibroblast growth factor-2 (FGF-2), transforming growth factor-beta (TGF-β), and insulin-like growth factor-1 (IGF-1), also improved the multi-lineage differentiation ability of stem cells [121,122,123]. Nandy et al. reported that FGF-2-treated hBM-MSCs showed the least cell death and the highest upregulation of tyrosine hydroxylase, a dopaminergic neuron marker, compared to other growth-factor-treated cells [121]. Meanwhile, Longobardi et al. reported that the addition of TGF-β and IGF-1 exerted both proliferative and anti-apoptotic actions and even induced the differentiation of BM-MSCs into chondrocytes, showing an increase in the expression of chondrogenic markers in mice [122]. In addition, IGF-1 stimulates osteoblastic differentiation by activating the mTOR signalling pathway and helps maintain bone marrow mass in mice and rats [123]. The effects of substances for improving the efficiency of BM-MSC therapy are summarized in Table 1. Hence, not only the removal of harmful factors such as ROS but also the addition of supplements can be helpful for maintaining hBM-MSC stemness.

### 3.2. PBS Storage Time Is a Critical Factor for the Differentiation of hBM-MSCs in Gene Expression and Amino Acid Levels

For transplantation, MSCs should be suspended in PBS to prevent inflammatory reactions caused by foreign proteins and serum [124]. To evaluate the effect of storage time on hBM-MSCs before transplantation, transcriptome profiling data of hBM-MSCs stored in PBS for 0, 6, and 12 h were acquired as described in our previous reports [57,58]. The transcriptomic changes in the 6 h- and 12 h-stored groups were compared to the 0 h-stored control groups using Ingenuity Pathway Analysis (IPA) web-based bioinformatics software (Qiagen, CA, USA) and analyzed. Genes that showed >1.5- and <-1.5-fold changes in expression levels were selected and used for transcriptomic analysis. In the 6 h-stored groups, 1466 genes showed such changes in their gene expression levels (806 downregulated and 660 upregulated), whereas in the 12 h-stored groups, 1817 genes showed such changes in their gene expression levels (1145 downregulated and 672 upregulated).

Computational prediction of cellular functions showed that the quantity of ROS, peroxidation of lipid, and cell damage were increased, and that the differentiation of stem cells was inhibited in both 6 h- and 12 h-stored hBM-MSCs compared to the control (Figure 2). Moreover, these transcriptomic networks showed that ROS levels, lipid peroxidation, cellular damage, and differentiation abilities were related. Interestingly, predictions of all these cellular functions were more robustly represented in the 12 h-storage groups than in the 6 h-storage groups (Figure 2B), suggesting that ROS production, lipid peroxidation, cell damage, and stem cell differentiation were adversely affected by increasing the PBS storage time in hBM-MSCs. Detailed fold-change information for each gene in the transcriptomic networks is listed in Table 2 (6 h-stored groups) and Table 3 (12 h-stored groups). Further experimental studies are required to elucidate the differentiation ability of hBM-MSCs.

In the omics analysis approaches, each analysis of single omics has advantages and disadvantages. For example, transcriptomics includes tremendous gene expression data, but it is not able to reflect the exact phenotype of the organism [125]. Metabolomics shows the “end products” of biological processes and reflects more exact phenotypes. However, the metabolites that can be analyzed are limited by experimental methods [126]. Hence, an integrated omics approach can compensate for the shortcomings of each omics and provides a more comprehensive vision of complex biological processes. Indeed, using integrated omics approaches, the unexplained delicate toxicity of nanoparticles and particulate matter [65,127,128,129] was analyzed, and the biological meaning of mitochondrial diseases was found in various conditions [130].

Amino acid profiling omics data using GC-MS provided cellular information in hBM-MSCs over time [57,131], and the amino acid levels at each time point were integrated with transcriptomics to establish metabotranscriptomics which integrates transcriptomics and metabolomics. Fold changes in the levels of amino acids > 1.2 and <−1.2 were used, and nine and eight amino acids were integrated into the IPA networks of hBM-MSCs stored for 6 h and 12 h, respectively. In detail, in the 6 h-stored groups, GABA (2.289, fold change), glycine (−1.624), glutamine (1.702), lysine (1.214), phenylalanine (1.297), proline (−2.255), threonine (1.356), tyrosine (1.685), and pyrrolidonecarboxylic acid (−1.209) were used, and in the 12 h-stored groups, GABA (3.848), glycine (1.458), aspartic acid (−1.438), glutamic acid (1.358), lysine (4.572), serine (1.302), threonine (−1.799), and tyrosine (2.368) were used for analysis. The metabotranscriptomic networks showed a similar tendency to the transcriptomics prediction, maintaining a more robust prediction of cellular functions in the hBM-MSC group stored for 12 h (Figure 3). These transcriptomic and integrated omics analyses suggest that the PBS storage time is a crucial factor for preserving the cell quality and differentiation ability of hBM-MSCs.

Although our analyses have been reported previously [58], the data in the current review were analyzed using a new, well-developed bioinformatic program. Notably, the data-based program was curated by a data curator from previously published papers, and the results obtained using artificial intelligence are considered more objective biological information rather than data collected by individuals. Our bioinformatic study showing that the freshness of stem cells decreases over time is consistent with previous reports [56,57,58]. However, the differentiation ability of stem cells after cell detachment from culture dishes is inconsistent with in silico prediction and the osteogenic and adipogenic potentials in the previous report [57]. The discrepancies can be attributed to the experimental differences between omics data analyzed directly at each storage time point and long-term cell culture and frequent exchanges of culture medium for cell differentiation. In conclusion, the freshness and differentiation ability of stem cells in stem cell therapy are closely related to the time after cell detachment from culture dishes. In this regard, this bioinformatic study will be an essential factor to be considered in future stem cell therapies.

## 4. Conclusions

Herein, we review the application of hBM-MSCs to the treatment of various NDDs and analyze the quality and differentiation ability of hBM-MSCs detached from culture dishes using integrated omics analysis for successful stem cell therapy. Conventional analysis methods have limitations in analyzing the quality and differentiation ability of dissociated hBM-MSCs and cannot comprehensively elucidate the delicate relationships between the cell condition of dissociated hBM-MSCs and therapeutic effects. Multidisciplinary omics and integrated multi-omics approaches provide in-depth and comprehensive information on the quality characteristics and differentiation ability of dissociated hBM-MSCs. Because stemness is a complex process that combines proliferation and self-renewal to generate differentiated cells with identical genotypes, a fragmentary approach is not sufficient to refer to biological changes in the cell. In the future, comprehensive and computational analyses of dissociated hBM-MSCs at the genomic, transcriptomic, small RNAomic, proteomic, phenomic, and metabolomic levels using advanced machine learning algorithms will accelerate studies in the field of stem cell therapy. Thus, these approaches will be helpful in analyzing the condition of dissociated hBM-MSCs and improving their quality and differentiation ability for innovative and successful stem cell therapy.

## Figures and Tables

**Figure 1 bioengineering-10-00621-f001:**
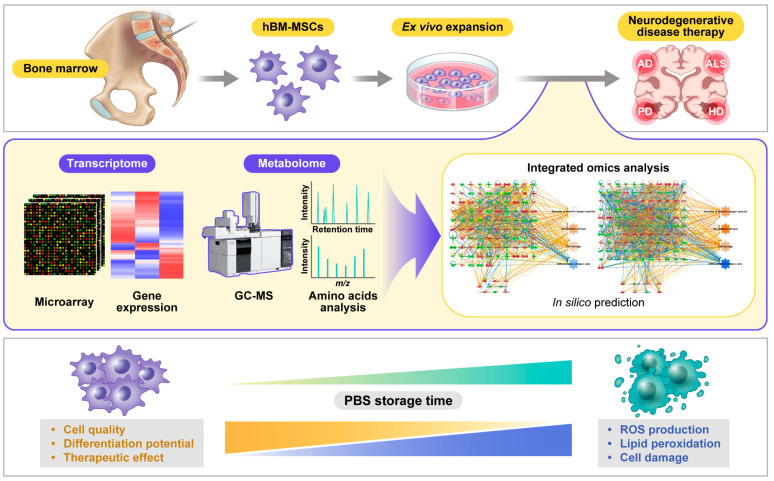
Effect of storage time on cell quality and differentiation potential. Before transplantation, human bone-marrow-derived mesenchymal stem cells (hBM-MSCs), which were expanded ex vivo and detached from the culture dish, were analyzed to evaluate their quality and differentiation potential. The transcriptome [58] and metabolome [57] were analyzed using microarray and gas chromatography-mass spectrometry (GC-MS), respectively. By integrating these datasets with in silico prediction, it was found that as the quality and differentiation potential of hBM-MSCs decreased and ROS production, lipid peroxidation, and cell damage increased in phosphate-buffered saline (PBS) over time.

**Figure 2 bioengineering-10-00621-f002:**
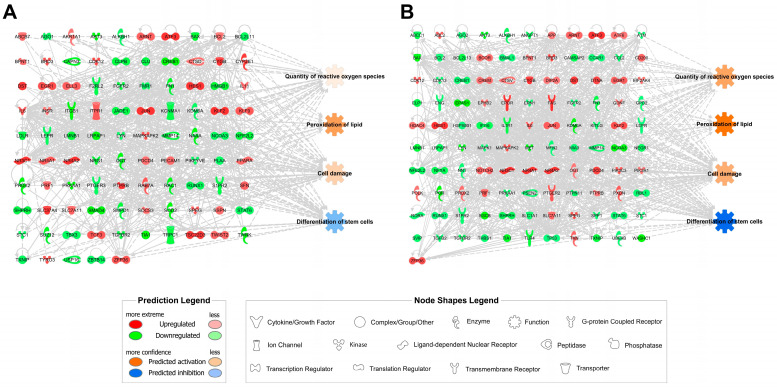
Transcriptomic networks with prediction using ingenuity pathway analysis. (**A**) Transcriptomic network in PBS-stored hBM-MSCs for 6 h; (**B**) Transcriptomic network in PBS-stored hBM-MSCs for 12 h. The fold change cut-off value used in the analysis for the networks is ±1.5 based on a previous report [58]. Upregulated genes are represented in red while the downregulated genes are in green. Orange and blue indicate the predicted activation and inhibition of cellular functions, respectively. Solid and dotted lines show direct and indirect relationships, respectively.

**Figure 3 bioengineering-10-00621-f003:**
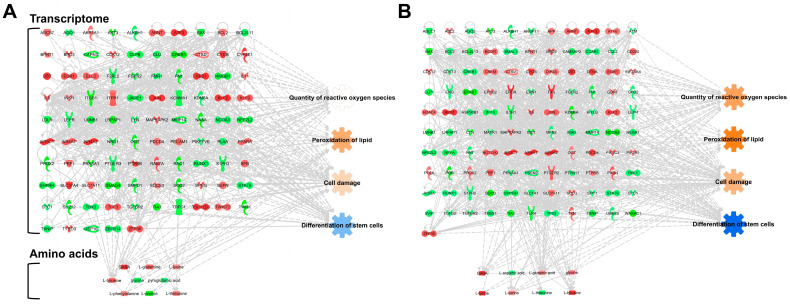
Metabotranscriptomic networks with prediction using ingenuity pathway analysis. (**A**) Metabotranscriptomic network in PBS-stored hBM-MSCs for 6 h; (**B**) metabotranscriptomic network in PBS-stored hBM-MSCs for 12 h. The fold change cut-off values used in the analysis for the networks are ±1.5 and ±1.2, respectively. Those used for transcriptome and amino acid analyses are based on a previous report [57]. The figure information is described in the legends of Figure 2.

**Table 1 bioengineering-10-00621-t001:** Summary of the functions and effects of substances for improving the efficiency of BM-MSC therapy.

Substances	Differentiation Direction	Functions and Effects	References
N-acetyl-L-cysteine (NAC)	Osteoblast	Decreased apoptosis Increased survival Increased GSH level Enhanced bone regeneration	[120]
Fibroblast growth factor-2 (FGF-2)	Dopaminergic neurons	Decreased cell death Increased upregulation of tyrosine hydroxylase Increased dopamine release	[121]
Transforming growth factor-beta (TGF-β)	Chondrocyte	Decreased apoptosis Increased cell proliferation Increased chondrogenic condensation and markers	[122]
Insulin-like growth factor-1 (IGF-1)	Chondrocyte	Decreased apoptosis Increased cell proliferation Increased chondrogenic condensation and markers	[122]
	Osteoblast	Induction of osteoblastic differentiation	[123]

**Table 2 bioengineering-10-00621-t002:** Ingenuity-pathway-analysis-based transcriptome profiles of the 6 h-stored hBM-MSCs.

Entrez Gene Name	Symbol	Entrez Gene ID ^a^	Location	Fold Change ^b^
ATP binding cassette subfamily B member 7	ABCB7	22	Cytoplasm	1.637
argonaute RISC component 1	AGO1	26,523	Cytoplasm	−1.542
aldo-keto reductase family 1 member A1	AKR1A1	10,327	Cytoplasm	1.512
AKT serine/threonine kinase 3	AKT3	10,000	Cytoplasm	−2.379
alkB homolog 1, histone H2A dioxygenase	ALKBH1	8846	Cytoplasm	−1.503
aryl hydrocarbon receptor nuclear translocator	ARNT	405	Nucleus	1.715
activating transcription factor 3	ATF3	467	Nucleus	2.88
BCL2 associated X, apoptosis regulator	BAX	581	Cytoplasm	−1.946
BCL2 apoptosis regulator	BCL2	596	Cytoplasm	1.534
BCL2 like 11	BCL2L11	10,018	Cytoplasm	−1.579
3′(2′), 5′-bisphosphate nucleotidase 1	BPNT1	10,380	Nucleus	1.564
bromodomain containing 3	BRD3	8019	Nucleus	1.855
calpain 3	CAPN3	825	Cytoplasm	−1.974
cyclin dependent kinase 12	CDK12	51,755	Nucleus	1.925
caseinolytic mitochondrial matrix peptidase chaperone subunit B	CLPB	81,570	Nucleus	−1.785
Clusterin	CLU	1191	Cytoplasm	−1.768
cAMP responsive element binding protein 1	CREB1	1385	Nucleus	−2.252
cathepsin D	CTSD	1509	Cytoplasm	1.527
Cytoglobin	CYGB	114,757	Cytoplasm	2.368
cytochrome P450 family 2 subfamily E member 1	CYP2E1	1571	Cytoplasm	2.095
Dystonin	DST	667	Plasma Membrane	2.126
early growth response 1	EGR1	1958	Nucleus	1.915
elongation factor for RNA polymerase II 3	ELL3	80,237	Nucleus	1.749
coagulation factor II thrombin receptor like 2	F2RL2	2151	Plasma Membrane	−1.714
fibroblast growth factor receptor 2	FGFR2	2263	Plasma Membrane	−1.859
fragile X messenger ribonucleoprotein 1	FMR1	2332	Cytoplasm	−1.569
fibronectin 1	FN1	2335	Extracellular Space	−1.951
hes family bHLH transcription factor 1	HES1	3280	Nucleus	8.134
high mobility group box 1	HMGB1	3146	Nucleus	−1.903
interleukin 11	IL11	3589	Extracellular Space	1.596
interleukin 6	IL6	3569	Extracellular Space	1.784
insulin receptor	INSR	3643	Plasma Membrane	1.561
integrin subunit beta 1	ITGB1	3688	Plasma Membrane	−2.31
inositol 1,4,5-trisphosphate receptor type 1	ITPR1	3708	Cytoplasm	1.775
jade family PHD finger 1	JADE1	79,960	Nucleus	−1.76
Jun proto-oncogene, AP-1 transcription factor subunit	JUN	3725	Nucleus	2.212
potassium calcium-activated channel subfamily M alpha 1	KCNMA1	3778	Plasma Membrane	−1.726
lysine demethylase 6A	KDM6A	7403	Nucleus	−1.511
KLF transcription factor 2	KLF2	10,365	Nucleus	2.261
KLF transcription factor 9	KLF9	687	Nucleus	1.803
low density lipoprotein receptor	LDLR	3949	Plasma Membrane	−1.783
leptin receptor	LEPR	3953	Plasma Membrane	−1.575
lamin B1	LMNB1	4001	Nucleus	−1.862
LDL receptor related protein associated protein 1	LRPAP1	4043	Plasma Membrane	−2.12
LYN proto-oncogene, Src family tyrosine kinase	LYN	4067	Cytoplasm	−1.541
MAPK activated protein kinase 2	MAPKAPK2	9261	Nucleus	1.848
matrix metallopeptidase 14	MMP14	4323	Extracellular Space	−1.874
N-acylethanolamine acid amidase	NAAA	27,163	Cytoplasm	−2.184
nuclear receptor coactivator 3	NCOA3	8202	Nucleus	−1.572
NFE2 like bZIP transcription factor 2	NFE2L2	4780	Nucleus	−1.823
nuclear receptor subfamily 3 group C member 1	NR3C1	2908	Nucleus	2.214
nuclear receptor subfamily 4 group A member 1	NR4A1	3164	Nucleus	1.808
nuclear receptor subfamily 4 group A member 2	NR4A2	4929	Nucleus	4.047
neuregulin 1	NRG1	3084	Plasma Membrane	−1.913
O-linked N-acetylglucosamine (GlcNAc) transferase	OGT	8473	Cytoplasm	−2.058
programmed cell death 4	PDCD4	27,250	Nucleus	1.554
platelet and endothelial cell adhesion molecule 1	PECAM1	5175	Plasma Membrane	1.602
phosphoinositide kinase, FYVE-type zinc finger containing	PIKFYVE	200,576	Cytoplasm	−1.529
phospholipase A2 activating protein	PLAA	9373	Cytoplasm	−1.507
peroxisome proliferator activated receptor alpha	PPARA	5465	Nucleus	1.646
peroxiredoxin 2	PRDX2	7001	Cytoplasm	−2.276
perforin 1	PRF1	5551	Cytoplasm	1.668
protein kinase AMP-activated catalytic subunit alpha 1	PRKAA1	5562	Cytoplasm	−2.252
prostaglandin E receptor 3	PTGER3	5733	Plasma Membrane	−1.539
protein tyrosine phosphatase receptor type B	PTPRB	5787	Plasma Membrane	2.444
RAB7A, member RAS oncogene family	RAB7A	7879	Cytoplasm	1.801
Rac family small GTPase 1	RAC1	5879	Plasma Membrane	−2.019
RUNX family transcription factor 1	RUNX1	861	Nucleus	−1.616
sphingosine-1-phosphate receptor 2	S1PR2	9294	Plasma Membrane	−1.73
Stratifin	SFN	2810	Cytoplasm	1.7
SNF2 histone linker PHD RING helicase	SHPRH	257,218	Nucleus	−1.812
solute carrier family 37 member 4	SLC37A4	2542	Cytoplasm	3.046
solute carrier family 7 member 11	SLC7A11	23,657	Plasma Membrane	1.721
SMAD family member 4	SMAD4	4089	Nucleus	−2.153
sphingomyelin phosphodiesterase 1	SMPD1	6609	Cytoplasm	−1.834
suppressor of cytokine signaling 3	SOCS3	9021	Cytoplasm	1.628
superoxide dismutase 2	SOD2	6648	Cytoplasm	−2.242
striated muscle enriched protein kinase	SPEG	10,290	Nucleus	1.771
Sarcospan	SSPN	8082	Plasma Membrane	1.575
signal transducer and activator of transcription 6	STAT6	6778	Nucleus	−1.571
stanniocalcin 1	STC1	6781	Extracellular Space	−1.567
SUZ12 polycomb repressive complex 2 subunit	SUZ12	23,512	Nucleus	−2.001
T-box transcription factor 3	TBX3	6926	Nucleus	−1.533
transcription factor 3	TCF3	6929	Nucleus	1.81
transforming growth factor beta receptor 2	TGFBR2	7048	Plasma Membrane	−1.822
TIA1 cytotoxic granule associated RNA binding protein	TIA1	7072	Nucleus	−2.369
transient receptor potential cation channel subfamily C member 1	TRPC1	7220	Plasma Membrane	−1.812
TSC22 domain family member 3	TSC22D3	1831	Nucleus	2.134
twist family bHLH transcription factor 2	TWIST2	117,581	Nucleus	1.652
twinkle mtDNA helicase	TWNK	56,652	Cytoplasm	−2.459
thioredoxin interacting protein	TXNIP	10,628	Cytoplasm	−1.567
TYRO3 protein tyrosine kinase	TYRO3	7301	Plasma Membrane	1.905
ubiquitin specific peptidase 16	USP16	10,600	Cytoplasm	−1.764
zinc finger and BTB domain containing 14	ZBTB14	7541	Nucleus	−1.55
ZFP36 ring finger protein	ZFP36	7538	Nucleus	1.827

^a^ The Entrez gene ID is the unique integer identifier for humans; ^b^ normalized signal fold change in the 6 h-stored group to the corresponding signal of the control group.

**Table 3 bioengineering-10-00621-t003:** Ingenuity-pathway-analysis-based transcriptome profiles of the 12 h-stored hBM-MSCs.

Entrez Gene Name	Symbol	Entrez Gene ID ^a^	Location	Fold Change ^b^
ATP binding cassette subfamily C member 1	ABCC1	4363	Plasma Membrane	−1.984
ABL proto-oncogene 2, non-receptor tyrosine kinase	ABL2	27	Cytoplasm	1.744
argonaute RISC catalytic component 2	AGO2	27,161	Cytoplasm	−1.588
AKT serine/threonine kinase 3	AKT3	10,000	Cytoplasm	−2.393
alkB homolog 1, histone H2A dioxygenase	ALKBH1	8846	Cytoplasm	−1.83
angiopoietin 1	ANGPT1	284	Extracellular Space	−1.869
amyloid beta precursor protein	APP	351	Plasma Membrane	1.506
aryl hydrocarbon receptor nuclear translocator	ARNT	405	Nucleus	1.749
activating transcription factor 3	ATF3	467	Nucleus	2.535
activating transcription factor 6	ATF6	22,926	Cytoplasm	1.56
ATM serine/threonine kinase	ATM	472	Nucleus	−1.523
BCL2 associated X, apoptosis regulator	BAX	581	Cytoplasm	−2.51
BCL2 apoptosis regulator	BCL2	596	Cytoplasm	−1.646
BCL2 like 13	BCL2L13	23,786	Cytoplasm	−2.078
BCL6 corepressor	BCOR	54,880	Nucleus	1.656
basic helix-loop-helix ARNT like 1	BMAL1	406	Nucleus	−1.562
3′(2′), 5′-bisphosphate nucleotidase 1	BPNT1	10,380	Nucleus	1.565
bromodomain containing 3	BRD3	8019	Nucleus	2.089
calmodulin regulated spectrin associated protein family member 2	CAMSAP2	23,271	Cytoplasm	−1.955
cell division cycle and apoptosis regulator 1	CCAR1	55,749	Nucleus	−1.579
C-C motif chemokine ligand 2	CCL2	6347	Extracellular Space	−1.655
CD200 molecule	CD200	4345	Plasma Membrane	1.713
cyclin dependent kinase 12	CDK12	51,755	Nucleus	1.849
cyclin dependent kinase 13	CDK13	8621	Nucleus	−1.555
cAMP responsive element binding protein 1	CREB1	1385	Nucleus	−1.68
cAMP responsive element modulator	CREM	1390	Nucleus	1.601
cathepsin V	CTSV	1515	Cytoplasm	1.573
cytoglobin	CYGB	114,757	Cytoplasm	2.378
disco interacting protein 2 homolog A	DIP2A	23,181	Nucleus	1.549
dystonin	DST	667	Plasma Membrane	2.201
dystrobrevin alpha	DTNA	1837	Plasma Membrane	2.069
early growth response 1	EGR1	1958	Nucleus	1.786
eukaryotic translation initiation factor 2 alpha kinase 4	EIF2AK4	440,275	Cytoplasm	1.576
elongator acetyltransferase complex subunit 1	ELP1	8518	Cytoplasm	−2.024
endoglin	ENG	2022	Plasma Membrane	−1.629
endothelial PAS domain protein 1	EPAS1	2034	Nucleus	−3.025
EPH receptor B2	EPHB2	2048	Plasma Membrane	1.591
erythropoietin receptor	EPOR	2057	Plasma Membrane	2.778
endoplasmic reticulum to nucleus signaling 1	ERN1	2081	Cytoplasm	−1.588
Fas cell surface death receptor	FAS	355	Plasma Membrane	2.388
fibroblast growth factor receptor 2	FGFR2	2263	Plasma Membrane	−1.515
fibronectin 1	FN1	2335	Extracellular Space	−2.059
glial cell derived neurotrophic factor	GDNF	2668	Extracellular Space	1.512
glycerol-3-phosphate dehydrogenase 2	GPD2	2820	Cytoplasm	−1.54
histone deacetylase 4	HDAC4	9759	Nucleus	1.563
hes family bHLH transcription factor 1	HES1	3280	Nucleus	5.645
heat shock protein 90 beta family member 1	HSP90B1	7184	Cytoplasm	−1.51
interferon gamma inducible protein 16	IFI16	3428	Nucleus	−1.892
interleukin 1 receptor type 1	IL1R1	3554	Plasma Membrane	−2.05
interleukin 6	IL6	3569	Extracellular Space	1.777
Jun proto-oncogene, AP-1 transcription factor subunit	JUN	3725	Nucleus	1.827
lysine demethylase 6A	KDM6A	7403	Nucleus	−2.947
KIT ligand	KITLG	4254	Extracellular Space	−1.654
KLF transcription factor 2	KLF2	10,365	Nucleus	1.979
leptin receptor	LEPR	3953	Plasma Membrane	−1.507
lamin B1	LMNB1	4001	Nucleus	−2.167
LDL receptor related protein associated protein 1	LRPAP1	4043	Plasma Membrane	−1.845
LYN proto-oncogene, Src family tyrosine kinase	LYN	4067	Cytoplasm	−2.418
mitogen-activated protein kinase 1	MAPK1	5594	Cytoplasm	−1.564
MAPK activated protein kinase 2	MAPKAPK2	9261	Nucleus	1.95
MET proto-oncogene, receptor tyrosine kinase	MET	4233	Plasma Membrane	−2.447
mitofusin 2	MFN2	9927	Cytoplasm	−1.554
MIA SH3 domain ER export factor 3	MIA3	375,056	Cytoplasm	−1.621
matrix metallopeptidase 14	MMP14	4323	Extracellular Space	−1.7
nuclear receptor coactivator 3	NCOA3	8202	Nucleus	−2.462
neuronal growth regulator 1	NEGR1	257,194	Plasma Membrane	−1.799
NFE2 like bZIP transcription factor 2	NFE2L2	4780	Nucleus	−1.85
nuclear transcription factor Y subunit alpha	NFYA	4800	Nucleus	−1.914
nicotinamide nucleotide transhydrogenase	NNT	23,530	Cytoplasm	−1.592
notch receptor 2	NOTCH2	4853	Plasma Membrane	1.741
nuclear receptor subfamily 3 group C member 1	NR3C1	2908	Nucleus	2.526
nuclear receptor subfamily 4 group A member 1	NR4A1	3164	Nucleus	2.282
nuclear receptor subfamily 4 group A member 2	NR4A2	4929	Nucleus	5.217
O-linked N-acetylglucosamine (GlcNAc) transferase	OGT	8473	Cytoplasm	1.511
programmed cell death 4	PDCD4	27,250	Nucleus	2.085
phosphatidylinositol 3-kinase catalytic subunit type 3	PIK3C3	5289	Cytoplasm	1.887
phosphoinositide-3-kinase regulatory subunit 1	PIK3R1	5295	Cytoplasm	1.557
DNA polymerase kappa	POLK	51,426	Nucleus	1.562
cytochrome p450 oxidoreductase	POR	5447	Cytoplasm	−2.999
peroxiredoxin 2	PRDX2	7001	Cytoplasm	1.62
perforin 1	PRF1	5551	Cytoplasm	1.888
protein kinase AMP-activated catalytic subunit alpha 1	PRKAA1	5562	Cytoplasm	−1.683
presenilin 2	PSEN2	5664	Cytoplasm	−1.611
prostaglandin E receptor 2	PTGER2	5732	Plasma Membrane	1.877
protein tyrosine phosphatase non-receptor type 11	PTPN11	5781	Cytoplasm	−1.714
protein tyrosine phosphatase receptor type B	PTPRB	5787	Plasma Membrane	1.544
peroxidasin	PXDN	7837	Extracellular Space	1.827
RB transcriptional corepressor like 1	RBL1	5933	Nucleus	−1.614
RAR related orphan receptor A	RORA	6095	Nucleus	−1.609
RUNX family transcription factor 1	RUNX1	861	Nucleus	−1.563
sphingosine-1-phosphate receptor 2	S1PR2	9294	Plasma Membrane	−1.619
sarcoglycan beta	SGCB	6443	Plasma Membrane	−2.825
SNF2 histone linker PHD RING helicase	SHPRH	257,218	Nucleus	−1.917
solute carrier family 1 member 1	SLC1A1	6505	Plasma Membrane	−1.692
solute carrier family 7 member 11	SLC7A11	23,657	Plasma Membrane	1.719
striated muscle enriched protein kinase	SPEG	10,290	Nucleus	1.96
secreted phosphoprotein 1	SPP1	6696	Extracellular Space	−1.573
signal transducer and activator of transcription 6	STAT6	6778	Nucleus	−1.638
stanniocalcin 1	STC1	6781	Extracellular Space	−1.546
small VCP interacting protein	SVIP	258,010	Cytoplasm	−1.696
transforming growth factor beta 2	TGFB2	7042	Extracellular Space	−1.584
transforming growth factor beta receptor 2	TGFBR2	7048	Plasma Membrane	−1.801
thrombospondin 1	THBS1	7057	Extracellular Space	−1.66
TIA1 cytotoxic granule associated RNA binding protein	TIA1	7072	Nucleus	−2.305
toll like receptor 4	TLR4	7099	Plasma Membrane	−2.167
tumor protein p53	TP53	7157	Nucleus	−1.518
thioredoxin	TXN	7295	Cytoplasm	1.692
thioredoxin interacting protein	TXNIP	10,628	Cytoplasm	−2.163
ubiquitination factor E4B	UBE4B	10,277	Cytoplasm	−1.591
WASH complex subunit 1	WASHC1	100,287,171	Cytoplasm	−3.587
ZFP36 ring finger protein	ZFP36	7538	Nucleus	1.942

^a^ The Entrez gene ID is the unique integer identifier for humans; ^b^ normalized signal fold change in the 12 h-stored group to the corresponding signal of the control group.

## Data Availability

The data used to support the findings are included within the article.

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
