# Peer review of "Human Bone Marrow-Derived Mesenchymal Stem Cell Applications in Neurodegenerative Disease Treatment and Integrated Omics Analysis for Successful Stem Cell Therapy"

_bioengineering, 2023, doi:10.3390/bioengineering10050621_

Round 1

Reviewer 1 Report

This is a very meaningful article. The idea in this manuscript is novel. This is a carefully done study and the findings are of considerable interest. My detailed comments are as follows

a.Please compare BMSCs with other souce of MSCs, such as adipose umbilical cord, and urine.

b.Please explain the effect of age or gender on BMSCs.

c. The author need to add the challenge part in the manuscript.

d. Currently, many biomaterials combined with BMSCs for tissue repair, please add some content about biomaterials combined with BMSCs for tissue repair.

Author Response

Answers to the comments of Reviewer #1

This is a very meaningful article. The idea in this manuscript is novel. This is a carefully done study and the findings are of considerable interest. My detailed comments are as follows:

a. Please compare BMSCs with other source of MSCs, such as adipose umbilical cord, and urine.

> As per your comment, we added the bone marrow-derived mesenchymal stem cell’s specific advantages compared to other sources of MSCs or other stem cells as follows:

Line 62-68, Page 2

For these reasons, BM-MSCs are extensively used in clinical applications and therefore the biological characteristics and clinical effects of BM-MSCs are far more well established than MSCs obtained from other sources [23]. In addition, the expression level of HLA class I or II is low in BM-MSCs, avoiding activation of allogenic lymphocytes [24], and BM-MSCs are safe from tumorigenicity after transplantation than other stem cells, including induced pluripotent stem cells (iPSCs) and neural stem cells [25,26].

Related References

  1. Musial-Wysocka, A.; Kot, M.; Majka, M. The Pros and Cons of Mesenchymal Stem Cell-Based Therapies. Cell Transplant. 2019, 28, 801-812.
  2. Klimczak, A.; Kozlowska, U.; Kurpisz, M. Muscle Stem/Progenitor Cells and Mesenchymal Stem Cells of Bone Marrow Origin for Skeletal Muscle Regeneration in Muscular Dystrophies. Arch Immunol Ther Exp (Warsz). 2018, 66, 341-354.
  3. Wang, Y.H.; Wang, D.R.; Guo, Y.C.; Liu, J.Y.; Pan, J. The application of bone marrow mesenchymal stem cells and biomaterials in skeletal muscle regeneration. Regen Ther. 2020, 15, 285-294.
  4. Xu, L.; Zhang, M.; Shi, L.; Yang, X.; Chen, L.; Cao, N.; Lei, A.; Cao, Y. Neural stemness contributes to cell tumorigenicity. Cell Biosci. 2021, 11, 21.

b. Please explain the effect of age or gender on BMSCs.

> According to your comment, we added the effect of gender on BMSCs. However, regarding the age, we already mentioned it in the main text, please check Line 291-301, Page 7.

Line 267-276, Page 6

Meanwhile, there are some studies suggesting that gender may affect the efficiency of BM-MSC therapy [106,107]. Sammour et al. reported that female BM-MSCs have more therapeutic effects than male BM-MSCs by showing greater pro-angiogenic and anti-inflammatory effects in mice models [106]. In addition, Crisostomo et al. demonstrated that female BM-MSCs showed lower apoptosis, TNF and IL-6 production and higher VEGF expression upon stress activation than male, due to their inherent resistance to TNFR1 activation [107]. However, one study reported that in vitro mesodermal differential capacity of hBM-MSCs is not highly related to the donor gender [108]. Although donor gender seems to play a role in therapeutic effects of BM-MSCs, this is still not clearly elucidated and further studies are required to clarify the effect of gender on BM-MSCs.

Related References

  1. Sammour, I.; Somashekar, S.; Huang, J.; Batlahally, S.; Breton, M.; Valasaki, K.; Khan, A.; Wu, S.; Young, K.C. The Effect of Gender on Mesenchymal Stem Cell (MSC) Efficacy in Neonatal Hyperoxia-Induced Lung Injury. PLoS One. 2016, 11, e0164269.
  2. Crisostomo, P.R.; Wang, M.; Herring, C.M.; Markel, T.A.; Meldrum, K.K.; Lillemoe, K.D.; Meldrum, D.R. Gender differences in injury induced mesenchymal stem cell apoptosis and VEGF, TNF, IL-6 expression: role of the 55 kDa TNF receptor (TNFR1). J Mol Cell Cardiol. 2007, 42, 142-149.
  3. Siegel, G.; Kluba, T.; Hermanutz-Klein, U.; Bieback, K.; Northoff, H.; Schafer, R. Phenotype, donor age and gender affect function of human bone marrow-derived mesenchymal stromal cells. BMC Med. 2013, 11, 146.

c. The author need to add the challenge part in the manuscript.

> As you have pointed out, we added challenge part about our analyses as follows:

Line 380-393, Page 9

Although our analyses have been previously reported [58], data in the current review is analysed using a new well-developed bioinformatic program. Notably, the data-based program curated by a data curator from previously published papers, and the results obtained using artificial intelligence are considered more objective biological information rather than collecting data by individuals. Our bioinformatic study showing that the freshness of stem cells decreases over time is consistent with previous reports [56-58]. However, the differentiation ability of stem cells after cell detachment from culture dishes is inconsistent with in silico prediction and the osteogenic and adipogenic potentials in the previous report [57]. The discrepancies can be attributed to the experimental differences between omics data analysed directly at each storage time point, and long-term cell culture and frequent exchanges of culture medium for cell differentiation. In conclusion, the freshness and differentiation ability of stem cells in stem cell therapy are closely related to the time after cell detachment from culture dishes. In this regard, this bioinformatic study will be an essential factor to be considered in future stem cell therapies.

Related References

  1. Lee, D.Y.; Lee, S.E.; Kwon, D.H.; Nithiyanandam, S.; Lee, M.H.; Hwang, J.S.; Basith, S.; Ahn, J.H.; Shin, T.H.; Lee, G. Strategies to Improve the Quality and Freshness of Human Bone Marrow-Derived Mesenchymal Stem Cells for Neurological Diseases. Stem Cells Int. 2021, 2021, 8444599.
  2. Shin, T.H.; Lee, S.; Choi, K.R.; Lee, D.Y.; Kim, Y.; Paik, M.J.; Seo, C.; Kang, S.; Jin, M.S.; Yoo, T.H.; et al. Quality and freshness of human bone marrow-derived mesenchymal stem cells decrease over time after trypsinization and storage in phosphate-buffered saline. Sci Rep. 2017, 7, 1106.
  3. Lee, K.A.; Shim, W.; Paik, M.J.; Lee, S.C.; Shin, J.Y.; Ahn, Y.H.; Park, K.; Kim, J.H.; Choi, S.; Lee, G. Analysis of changes in the viability and gene expression profiles of human mesenchymal stromal cells over time. Cytotherapy. 2009, 11, 688-697.

d. Currently, many biomaterials combined with BMSCs for tissue repair, please add some content about biomaterials combined with BMSCs for tissue repair.

> According to your comment, we described the MSCs-based tissue engineering combined with biomaterials as follows:

Line 75-97, Page 2

Because MSCs can accelerate tissue regeneration, they are excellent candidates for tissue engineering. Vacanti and Langer define tissue engineering as an interdisciplinary field in which engineering and biological sciences are applied together to develop biological substitutes that can restore, maintain, or enhance tissue function [51]. Tissue engineering facilitates autologous MSCs transplantation by seeding the patient’s cells onto a biodegradable scaffold that forms a specific tissue that can then be used to repair injuries [52]. Despite all this, MSC-based tissue engineering faces some challenges, such as the low survival rates of MSCs and the uncertainty of MSC differentiation after infusion. To address these shortcomings, biomaterials are used together with MSCs to maintain their viability, serve as a substrate for cell adhesion, induce differentiation into specific target cells, and as a mechanical tool for tissue regeneration. Biomaterials have demonstrated excellent biocompatibility, provide a suitable cellular microenvironment for MSCs, and are effective and easy to administer. As biomaterials are diverse in their physical, chemical, mechanical, and biological properties, they can contribute to tissue regeneration in different types of injuries [53]. Although several biomaterials have been developed, hydrogels, nanofibers, carbon-based nanomaterials, and cell-free scaffolds have emerged as frontrunners. For example, 3D structures of the hydrogel support proliferation of cells while acting as a barrier to harmful factors [54]. Yan et al. investigated collagen-chitosan scaffolds with BM-MSCs as a therapeutic strategy for traumatic brain injury. Collagen scaffolds with human MSCs have been shown to improve spatial learning and sensorimotor functions, while chitosan serves as a neuroprotector. They reported that these scaffolds exhibited low immunogenicity, good biocompatibility, and therapeutic effects such as neurological, behavioural, and cognitive recovery [55].

Related References

  1. Vacanti, J.P.; Langer, R. Tissue engineering: the design and fabrication of living replacement devices for surgical reconstruction and transplantation. Lancet. 1999, 354, S32-S34.
  2. Kassem, M.; Abdallah, B.M. Human bone-marrow-derived mesenchymal stem cells: biological characteristics and potential role in therapy of degenerative diseases. Cell Tissue Res. 2008, 331, 157-163.
  3. Li, J.; Liu, Y.; Zhang, Y.; Yao, B.; Enhejirigala; Li, Z.; Song, W.; Wang, Y.; Duan, X.; Yuan, X.; et al. Biophysical and Biochemical Cues of Biomaterials Guide Mesenchymal Stem Cell Behaviors. Front Cell Dev Biol. 2021, 9, 640388.
  4. Fernández-Serra, R.; Gallego, R.; Lozano, P.; González-Nieto, D. Hydrogels for neuroprotection and functional rewiring: a new era for brain engineering. Neural Regen Res. 2020, 15, 783.
  5. Yan, F.; Li, M.; Zhang, H.-Q.; Li, G.-L.; Hua, Y.; Shen, Y.; Ji, X.-M.; Wu, C.-J.; An, H.; Ren, M. Collagen-chitosan scaffold impregnated with bone marrow mesenchymal stem cells for treatment of traumatic brain injury. Neural Regen Res. 2019, 14, 1780.

Reviewer 2 Report

The authors discuss application hBM-MSCs as an approach of stem cell therapy in neurodegenerative disease. They also attempt to present integrated omics for characterization of hBM-MSC quality and differentiation ability during PBS storage. This topic is interesting but the review needs significant improvements before it can be considered for publication in Bioengineering.

1. The second section discussing integrated omics analyses of hBM-MSC quality and differentiation ability is mainly based on authors’ own work. In particular, tables 1 and 2 along with some descriptions of metabotranscriptomics changes in the text resemble research results and are not relevant in the review. In addition, although the authors conclude that bioinformatic data are consistent with experimental analyses, they do not provide a detailed discussion of literature on how PBS storage period affects differentiation ability and transplantation efficiency of hBM-MSCs.

2. Similarly, how integrated omics analyses of cell characteristics improve successful stem cell therapy is not clearly discussed. The authors may provide a few examples if these are available in the literature.

3. The authors summarize several methods for improving the efficiency of hBM-MSC therapy while preserving cell stemness. This is interesting but it would help the readers if they present a thorough list of these methods in the form of a table.

4. The authors may consider shortening the very wordy article title at least by deleting a few words.

Author Response

Answers to the comments of Reviewer #2

The authors discuss application hBM-MSCs as an approach of stem cell therapy in neurodegenerative disease. They also attempt to present integrated omics for characterization of hBM-MSC quality and differentiation ability during PBS storage. This topic is interesting but the review needs significant improvements before it can be considered for publication in Bioengineering.

  1. The second section discussing integrated omics analyses of hBM-MSC quality and differentiation ability is mainly based on authors’ own work. In particular, tables 1 and 2 along with some descriptions of metabotranscriptomics changes in the text resemble research results and are not relevant in the review. In addition, although the authors conclude that bioinformatic data are consistent with experimental analyses, they do not provide a detailed discussion of literature on how PBS storage period affects differentiation ability and transplantation efficiency of hBM-MSCs.

> We are grateful for your comment that we missed in this paper and added response to your comment as:

Line 380-393, Page 9

Although our analyses have been previously reported [58], data in the current review is analysed using a new well-developed bioinformatic program. Notably, the data-based program curated by a data curator from previously published papers, and the results obtained using artificial intelligence are considered more objective biological information rather than collecting data by individuals. Our bioinformatic study showing that the freshness of stem cells decreases over time is consistent with previous reports [56-58]. However, the differentiation ability of stem cells after cell detachment from culture dishes is inconsistent with in silico prediction and the osteogenic and adipogenic potentials in the previous report [57]. The discrepancies can be attributed to the experimental differences between omics data analysed directly at each storage time point, and long-term cell culture and frequent exchanges of culture medium for cell differentiation. In conclusion, the freshness and differentiation ability of stem cells in stem cell therapy are closely related to the time after cell detachment from culture dishes. In this regard, this bioinformatic study will be an essential factor to be considered in future stem cell therapies.

Related References

  1. Lee, D.Y.; Lee, S.E.; Kwon, D.H.; Nithiyanandam, S.; Lee, M.H.; Hwang, J.S.; Basith, S.; Ahn, J.H.; Shin, T.H.; Lee, G. Strategies to Improve the Quality and Freshness of Human Bone Marrow-Derived Mesenchymal Stem Cells for Neurological Diseases. Stem Cells Int. 2021, 2021, 8444599.
  2. Shin, T.H.; Lee, S.; Choi, K.R.; Lee, D.Y.; Kim, Y.; Paik, M.J.; Seo, C.; Kang, S.; Jin, M.S.; Yoo, T.H.; et al. Quality and freshness of human bone marrow-derived mesenchymal stem cells decrease over time after trypsinization and storage in phosphate-buffered saline. Sci Rep. 2017, 7, 1106.
  3. Lee, K.A.; Shim, W.; Paik, M.J.; Lee, S.C.; Shin, J.Y.; Ahn, Y.H.; Park, K.; Kim, J.H.; Choi, S.; Lee, G. Analysis of changes in the viability and gene expression profiles of human mesenchymal stromal cells over time. Cytotherapy. 2009, 11, 688-697.

  1. Similarly, how integrated omics analyses of cell characteristics improve successful stem cell therapy is not clearly discussed. The authors may provide a few examples if these are available in the literature.

> As per your comment, we added the contents about limitations of each single omics analysis, advantages of integrated omics analysis, and examples of integrated omics analysis as follows:

Line 355-364, Page 8

In the omics analysis approaches, each analysis of single omics has advantages and disadvantages. For example, transcriptomics includes tremendous gene expression data, but it is not able to reflect the exact phenotype of the organism [125]. Metabolomics shows the “end products” of biological processes and reflects more exact phenotypes. However, metabolites that can be analysed are limited by experimental methods [126]. Hence, an integrated omics approach can compensate for the shortcomings of each omics and provides a more comprehensive vision of complex biological processes. Indeed, using integrated omics approaches, the unexplained delicate toxicity of nanoparticles and particulate matter [65,127-129] was analysed, and the biological meaning of mitochondrial diseases were found in various conditions [130].

Related References

  1. Shin, T.H.; Manavalan, B.; Lee, D.Y.; Basith, S.; Seo, C.; Paik, M.J.; Kim, S.W.; Seo, H.; Lee, J.Y.; Kim, J.Y.; et al. Silica-coated magnetic-nanoparticle-induced cytotoxicity is reduced in microglia by glutathione and citrate identified using integrated omics. Part Fibre Toxicol. 2021, 18, 42.
  2. Karahalil, B. Overview of Systems Biology and Omics Technologies. Curr Med Chem. 2016, 23, 4221-4230.
  3. Johnson, C.H.; Gonzalez, F.J. Challenges and opportunities of metabolomics. J Cell Physiol. 2012, 227, 2975-2981.
  4. Shin, T.H.; Nithiyanandam, S.; Lee, D.Y.; Kwon, D.H.; Hwang, J.S.; Kim, S.G.; Jang, Y.E.; Basith, S.; Park, S.; Mo, J.S.; et al. Analysis of Nanotoxicity with Integrated Omics and Mechanobiology. Nanomaterials (Basel). 2021, 11, 2385.
  5. Shin, T.H.; Kim, S.G.; Ji, M.; Kwon, D.H.; Hwang, J.S.; George, N.P.; Ergando, D.S.; Park, C.B.; Paik, M.J.; Lee, G. Diesel-derived PM(2.5) induces impairment of cardiac movement followed by mitochondria dysfunction in cardiomyocytes. Front Endocrinol (Lausanne). 2022, 13, 999475.
  6. Shim, W.; Paik, M.J.; Nguyen, D.T.; Lee, J.K.; Lee, Y.; Kim, J.H.; Shin, E.H.; Kang, J.S.; Jung, H.S.; Choi, S.; et al. Analysis of changes in gene expression and metabolic profiles induced by silica-coated magnetic nanoparticles. ACS Nano. 2012, 6, 7665-7680.
  7. Khan, S.; Ince-Dunn, G.; Suomalainen, A.; Elo, L.L. Integrative omics approaches provide biological and clinical insights: examples from mitochondrial diseases. J Clin Invest. 2020, 130, 20-28.

  1. The authors summarize several methods for improving the efficiency of hBM-MSC therapy while preserving cell stemness. This is interesting but it would help the readers if they present a thorough list of these methods in the form of a table.

> According to your advice, we summarized the methods for improving the efficiency of BM-MSC therapy while preserving cell stemness in Table 1.

Line 324-325, Page 8

The effects of substances for improving the efficiency of BM-MSC therapy are summarised in Table 1.

Page 7-8

Table 1. Summary of the functions and effects of substances for improving the efficiency of BM-MSC therapy.

Substances

Differentiation

direction

Functions and effects

References

N-acetyl-L-cysteine (NAC)

Osteoblast

Decreased apoptosis

Increased survival

Increased GSH level

Enhanced bone regeneration

[120]

Fibroblast growth factor-2 (FGF-2)

Dopaminergic neurons

Decreased cell death

Increased upregulation of tyrosine hydroxylase

Increased dopamine release

[121]

Transforming growth factor-beta (TGF-β)

Chondrocyte

Decreased apoptosis

Increased cell proliferation

Increased chondrogenic condensation and markers

[122]

Insulin-like growth factor-1 (IGF-1)

Chondrocyte

Decreased apoptosis

Increased cell proliferation

Increased chondrogenic condensation and markers

[122]

Osteoblast

Induction of osteoblastic differentiation

[123]

Related References

  1. Watanabe, J.; Yamada, M.; Niibe, K.; Zhang, M.; Kondo, T.; Ishibashi, M.; Egusa, H. Preconditioning of bone marrow-derived mesenchymal stem cells with N-acetyl-L-cysteine enhances bone regeneration via reinforced resistance to oxidative stress. Biomaterials. 2018, 185, 25-38.
  2. Nandy, S.B.; Mohanty, S.; Singh, M.; Behari, M.; Airan, B. Fibroblast Growth Factor-2 alone as an efficient inducer for differentiation of human bone marrow mesenchymal stem cells into dopaminergic neurons. J Biomed Sci. 2014, 21, 1-10.
  3. Longobardi, L.; O'Rear, L.; Aakula, S.; Johnstone, B.; Shimer, K.; Chytil, A.; Horton, W.A.; Moses, H.L.; Spagnoli, A. Effect of IGF‐I in the chondrogenesis of bone marrow mesenchymal stem cells in the presence or absence of TGF‐β signaling. J Bone Miner Res. 2006, 21, 626-636.
  4. Xian, L.; Wu, X.; Pang, L.; Lou, M.; Rosen, C.J.; Qiu, T.; Crane, J.; Frassica, F.; Zhang, L.; Rodriguez, J.P. Matrix IGF-1 maintains bone mass by activation of mTOR in mesenchymal stem cells. Nat Med. 2012, 18, 1095-1101.

  1. The authors may consider shortening the very wordy article title at least by deleting a few words.

> Agreeing with your comment, we modified the article title as follows:

Line 2-4, Page 1

Human bone marrow-derived mesenchymal stem cell applications in neurodegenerative disease treatment and integrated omics analysis for successful stem cell therapy

Round 2

Reviewer 2 Report

The revised manuscript has addressed issues raised in my previous review.